# Autologous skeletal myoblast patch implantation prevents the deterioration of myocardial ischemia and right heart dysfunction in a pressure-overloaded right heart porcine model

**Kanta Araki**[1], **Shigeru Miyagawa**[1], **Takuji Kawamura**[1], **Ryo Ishii**[2], **Tadashi Watabe**[3], **Akima Harada**[1], **Masaki Taira**[1], **Koichi Toda**[1], **Toru Kuratani**[1], **Takayoshi Ueno**[1], **Yoshiki Sawa**[1]*

**1** Department of Cardiovascular Surgery, Osaka University Graduate School of Medicine, Osaka, Japan, **2** Department of Pediatrics, Osaka University Graduate School of Medicine, Osaka, Japan, **3** Department of Nuclear Medicine and Tracer Kinetics, Osaka University Graduate School of Medicine, Osaka, Japan

* sawa-p@surg1.med.osaka-u.ac.jp

**Data Availability Statement:** All relevant data are within the manuscript and its Supporting Information files.

## Abstract

Right ventricular dysfunction is a predictor for worse outcomes in patients with congenital heart disease. Myocardial ischemia is primarily associated with right ventricular dysfunction in patients with congenital heart disease and may be a therapeutic target for right ventricular dysfunction. Previously, autologous skeletal myoblast patch therapy showed an angiogenic effect for left ventricular dysfunction through cytokine paracrine effects; however, its efficacy in right ventricular dysfunction has not been evaluated. Thus, this study aimed to evaluate the angiogenic effect of autologous skeletal myoblast patch therapy and amelioration of metabolic and functional dysfunction, in a pressure-overloaded right heart porcine model. Pulmonary artery stenosis was induced by a vascular occluder in minipigs; after two months, autologous skeletal myoblast patch implantation on the right ventricular free wall was performed (n = 6). The control minipigs underwent a sham operation (n = 6). The autologous skeletal myoblast patch therapy alleviated right ventricular dilatation and ameliorated right ventricular systolic and diastolic dysfunction. $^{11}$C-acetate kinetic analysis using positron emission tomography showed improvement in myocardial oxidative metabolism and myocardial flow reserve after cell patch implantation. On histopathology, a higher capillary density and vascular maturity with reduction of myocardial ischemia were observed after patch implantation. Furthermore, analysis of mRNA expression revealed that the angiogenic markers were upregulated, and ischemic markers were downregulated after patch implantation. Thus, autologous skeletal myoblast patch therapy ameliorated metabolic and functional dysfunction in a pressure-overloaded right heart porcine model, by alleviating myocardial ischemia through angiogenesis.

**Funding:** This study was funded by a grant from JSPS KAKENHI to KA (Grant No. 19K18177).

**Competing interests:** The authors have declared that no competing interests exist.

## Introduction

The prognosis of patients with congenital heart disease (CHD) is improving with advances in treatment strategies and interventions [1–3]. However, long-term cardiac overload owing to anatomical features and residual lesions after repair leads to heart failure [4–6], especially right heart failure [7,8], which is also known to be a predictor for poor outcomes [4,8–11]. Myocardial ischemia is primarily associated with right ventricular dysfunction, causing right heart failure in patients with CHD [4,8,12–20]. Thus, myocardial ischemia could be a potential therapeutic target for right heart failure. Specific treatments for right heart failure in patients with CHD are not well elucidated [8,21–24], and even ventricular assist device therapy and heart transplantation remain challenging treatments in patients with CHD and are not suitable for all of them [25–29].

In recent years, regenerative treatments such as cell therapy have gained increasing attention as newer treatment options for left heart failure and are progressively being integrated into clinical use [30,31]. Previously, we revealed the clinical efficacy of autologous skeletal myoblast patch transplantation therapy for left ventricular dysfunction associated with ischemic cardiomyopathy and dilated cardiomyopathy, through angiogenesis and anti-fibrosis, induced by cytokine paracrine effects in preclinical [32–37] and clinical studies [38–41]. These findings propose a possible efficacy of the same treatment for right ventricular dysfunction based on the ischemic etiology; however, this has not been evaluated before.

Hence, we hypothesized that autologous skeletal myoblast patch transplantation alleviated right ventricular dysfunction, and conducted a preclinical study using a pressure-overloaded right heart porcine model.

## Material and methods

All studies were performed after the approval of the ethics review committee for animal experimentation of Osaka University Graduate School of Medicine, Osaka, Japan (reference number: 30-056-005). All animal care procedures were conducted in compliance with the Principles of Laboratory Animal Care formulated by the National Society for Medical Research and the Guide for the Care and Use of Laboratory Animals prepared by the Institute of Animal Resources and published by the National Institutes of Health (publication no: 85–23, revised 1996).

### Preparation of animal models

Twelve Göttingen minipigs aged 5–6 months and weighing 10–12 kg (Oriental Yeast Corporation, Tokyo, Japan) were used in the experiments. The minipigs were anesthetized with an intravenous administration of ketamine (6 mg/kg) and sodium pentobarbital (10 mg/kg) for endotracheal intubation, and were maintained with inhaled isoflurane (1.5%–2%). The left intercostal space was opened to mount a vascular occluder (OA218-025; Unique Medical Co., Ltd., Tokyo, Japan) (Fig 1A) to the pulmonary artery trunk (Fig 1B). The pericardium was opened, and the vascular occluder was mounted to the pulmonary artery trunk and was connected to an access port (CP2AC-7Fr; Primetech Co., Ltd., Tokyo, Japan), which was implanted subcutaneously in the back. Pulmonary artery stenosis was gradually strengthened over a month to a stenosis velocity of over 3.0 m/s, evaluated using echocardiography by injecting 50% glycerin from the access port to inflate the vascular occluder. Pulmonary artery banding (PAB) was maintained at the same level of stenosis for the rest of the study period (Fig 1G). All the minipigs survived the induction of pulmonary artery stenosis, and two months after PAB, the degree of right ventricular dysfunction was checked using echocardiography and

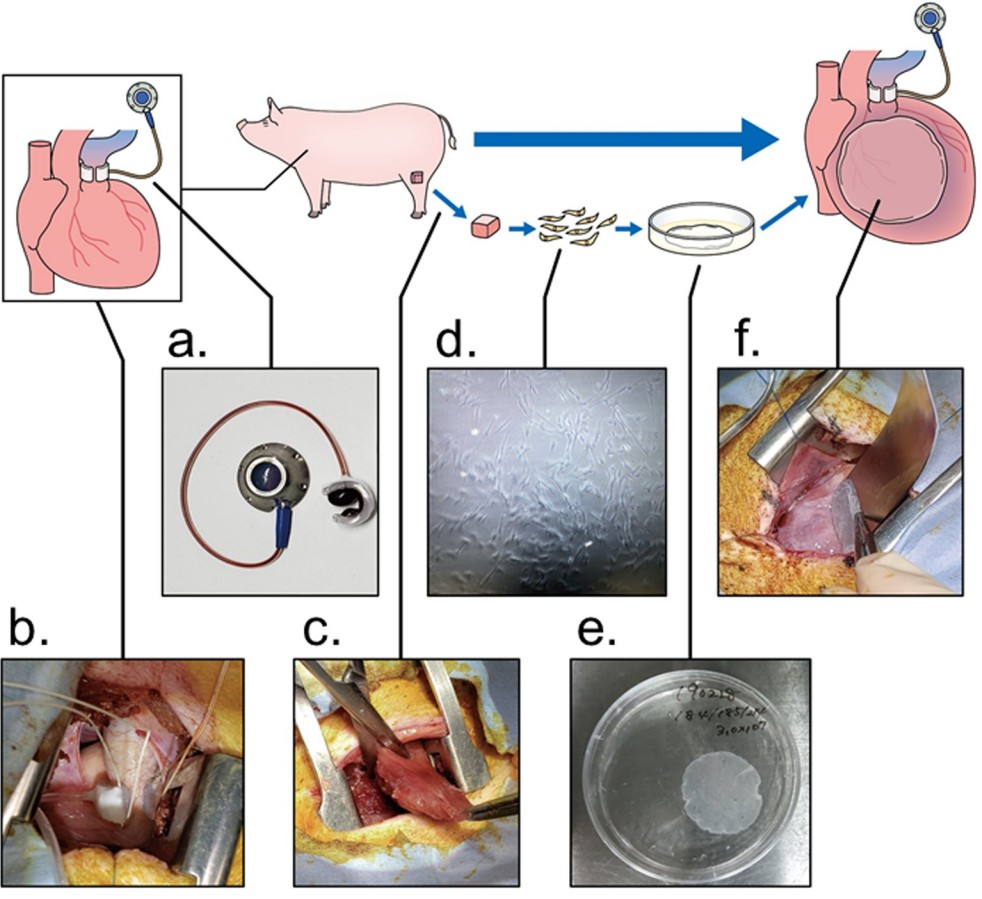

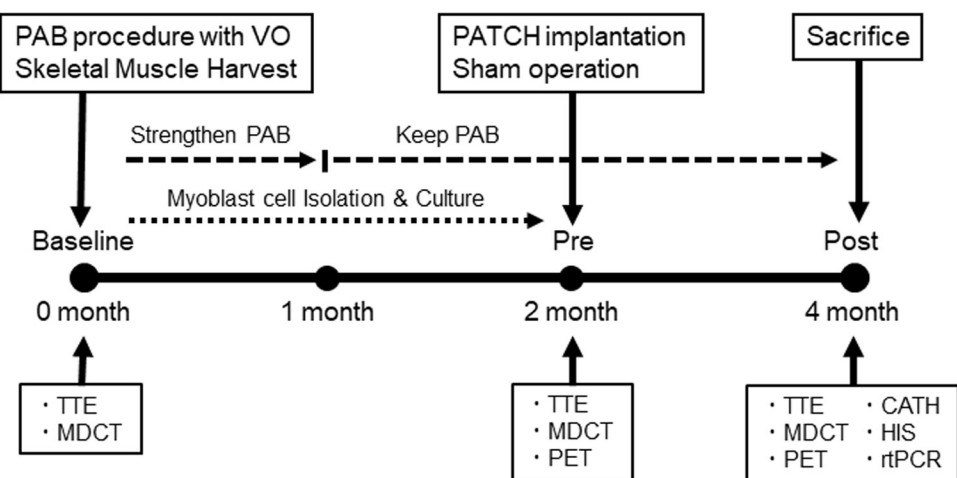

**Fig 1. Preparation of animal models and skeletal myoblast patch.** All minipigs underwent a PAB procedure by mounting the VO (a) to the pulmonary artery trunk (b) via the left intercostal space, harvesting the skeletal muscle from the quadriceps femoris muscle (c) at the same time. The purified skeletal myoblast cells were isolated and cultured (d). Cell patches were fabricated using a temperature-responsive dish (e) and were placed on the epicardium of the right ventricular free wall (f). The protocol of the study is given in (g). PAB, pulmonary artery banding; VO,

vascular occluder; TTE, transthoracic echocardiography; MDCT, electrocardiography-gated multidetector-row computed tomography; PET, positron emission tomography; CATH, catheterization; HIS, histologic analysis; rtPCR, real-time polymerase chain reaction analysis.

cardiac computed tomography, for confirming the establishment of a pressure-overloaded right heart failure model.

## Cell patch transplantation

Skeletal muscles weighing approximately 5 g were collected from the quadriceps femoris muscle during the PAB procedure (Fig 1C). Purified skeletal myoblast cells were cultured for three weeks as reported previously (Fig 1D) [35]. For quality control, the cell patch process was initiated after confirming that the cultured cells had a CD56 positivity rate of ≥80% using flow cytometry (BD FACSCanto2, BD Biosciences, New Jersey, USA). Cells were incubated in 100-mm temperature-responsive culture dishes (UpCell, CellSeed Inc., Tokyo, Japan) at 37°C for 6 h. The number of incubated cells was $5 \times 10^6$ per body weight of the minipigs, during transplantation. The dishes were then transferred to a clean bench at room temperature to release the cultured cells as intact cell patches. Using this protocol, skeletal myoblast cells spontaneously detached from the plates as free-floating monolayer cell patches (Fig 1E). Two months after the PAB procedure, minipigs were again placed under general anesthesia for either cell patch implantation (PATCH group, n = 6), or a sham operation with opening of the pericardium through the right intercostal space (control group, n = 6). The cell patch was placed on the epicardium of the right ventricular free wall (Fig 1F). Additionally, some fibrin glue was applied over the cell patches after their placement to attach them to the epicardium of the heart.

## Echocardiography

Before the PAB procedure, and at two months after PAB (that is, before the patch implantation or sham operation), and furthermore, at four months after PAB (that is, two months after the patch implantation or sham operation), transthoracic echocardiography (TTE) was performed to measure right ventricular systolic and diastolic function using Vivid I (General Electric Healthcare, Chicago, USA) with a 6Tc-RS probe (General Electric Healthcare, Chicago, USA) under general anesthesia. The following parameters were evaluated using four chamber view images and tissue Doppler images: (1) Tei index of right ventricle, (2) right ventricular fraction area change (RVFAC), (3) tricuspid annular plane systolic excursion (TAPSE), (4) the ratio between early tricuspid inflow velocity and early tricuspid annular diastolic velocity (E/e'), (5) isovolumic contraction time (ICT), (6) isovolumic relaxation time (IRT), and (7) right ventricular free wall thickness (RVFWT).

## Electrocardiography-gated multidetector-row computed tomography

Before the PAB procedure, and at two and four months after the PAB, electrocardiography-gated multidetector-row computed tomography (MDCT) was performed to measure the volume and contraction of the right ventricle. The images were taken in the supine position with a 16-slice multislice CT scanner (Somatron Emotion 16; Siemens Aktiengesellschaft, Munich, Germany) during end-expiratory breath hold, under general anesthesia. MDCT was performed after intravenous injection of a nonionic contrast medium (3 mL per bodyweight of the mini-pigs; Iomeprol; Bracco-Eisai Co Ltd, Tokyo, Japan). The axial images were reconstructed using the scanner software (Siemens Aktiengesellschaft, Munich, Germany). All

images were analyzed at a workstation (AZE; Virtual Pl Lexus 64, AZE Co., Ltd., Tokyo, Japan). Right ventricular end-diastolic volume (RVEDV) and right ventricular end-systolic volume (RVESV) were obtained from the workstation, and RV ejection fraction (RVEF) was calculated using the following equation: RVEF (%) = 100 × (RVEDV/RVESV)/(RVEDV).

## Pressure-volume loop analysis using cardiac catheterization

Four months after the PAB (that is, two months after the patch implantation or sham operation), an invasive pressure-volume loop analysis was performed under general anesthesia. A tourniquet was placed under the inferior vena cava to change the ventricular preload. Arterial access was obtained by introducing a 6Fr sheath into the left carotid artery, and venous access was obtained by introducing a 7Fr sheath into the right carotid vein. 4Fr conductance and pressure-tip catheters (CA-41063-PN, CD Leycom, Zoetermeer, Netherlands) were inserted into the left and right ventricles through the sheaths in the carotid artery and carotid vein, under fluoroscopic guidance. The position of the catheter was determined by observing the pressure and segmental volume signals with appropriate phase relationships. The conductance and pressure transducer controllers (Conduct NT Sigma 5DF plus analysis system; CFL-M, CD Leycom, Zoetermeer, Netherlands) were set, and pressure-volume loops and intracardiac electrocardiograms were monitored online. The conductance, pressure, and intracardiac electrocardiographic signals were analyzed with Inca software (CD Leycom, Zoetermeer, Netherlands). Under stable hemodynamic conditions, the baseline indices were initially measured, and the pressure-volume loop was then drawn during the inferior vena cava occlusion and analyzed. The following indices were calculated as the baseline left ventricular and right ventricular functions: heart rate (HR), ejection fraction (EF), end-systolic pressure (ESP), end-diastolic pressure (EDP), the maximum rate of the right ventricular pressure rise (dP/dt max), the maximal rate of fall of the right ventricular pressure (dP/dt min), and the time constant of isovolumic relaxation (Tau). The following relationships were determined by means of pressure-volume loop analysis, as load-independent measures of right ventricular function: end-systolic pressure-volume relationship (ESPVR) and end-diastolic pressure-volume relationship (EDPVR).

## $^{11}$C-acetate positron emission tomography

Two months after PAB (that is, before the patch implantation or sham operation), and four months after PAB (that is, two months after the patch implantation or sham operation), *in vivo* measurements of myocardial oxygen consumption (MVO$_2$) and myocardial blood flow (MBF) were performed using positron emission tomography (PET) scanning (Eminence-B SET-3000B/L; Shimadzu, Kyoto, Japan), on four minipigs in each group. The minipigs were sedated using a propofol infusion (1 mL/h), and with 1% isoflurane inhalation, and were positioned in a whole-body PET scanner at the Medical Imaging Center for Translational Research of Osaka University Graduate School of Medicine. A 5-min transmission scan was performed for correcting the emission images for photon attenuation. Immediately after the transmission scan, approximately 100 MBq of $^{11}$C-acetate was administered intravenously, and dynamic PET acquisition was initiated (15 mins). The measurements were performed under both resting conditions and high cardiac work state induced by catecholamine (dobutamine, 20 μg/kg per minute), as reported previously [42]. PET images were acquired in 30 frames (10 s × 6, 20 s × 6, 30 s × 12, and 60 s × 6) and were reconstructed with a dynamic row-action likelihood algorithm with an image matrix of 128 × 128. The PET images were analyzed using the software PMOD (Ver. 4.003, PMOD Japan Inc., Tokyo, Japan), where the right ventricle was also analyzed by flipping the PET image horizontally. To determine the global cardiac oxygen consumption, the acetate clearance rates (myocardial oxidative consumption [$k_{mono}$]) of all segments of each measurement were averaged. The MBF was

calculated using the one-tissue compartment model. Both the values of $k_{mono}$ and MBF were derived from the subtraction of the preoperative test value from the postoperative test value, to represent the amount of change by treatment. RV efficiency was derived by combining cardiac catheterization data, CT data, and RV weight using the following equation as reported previously [43]: RV efficiency = HR × RVESP × RVSV × $1.33 \times 10^{-4}$/RV $k_{mono}$ × RV mass × 20, where RVSV is the RV stroke volume determined using CT, RVESP is the RV end-systolic pressure determined using cardiac catheterization, HR is heart rate determined using cardiac catheterization, RV $k_{mono}$ is the resting condition right ventricular $k_{mono}$ at two months after patch implantation or sham operation, and RV mass is the RV weight of the harvested heart.

## Histologic analyses

Four months after PAB (that is, two months after patch implantation or sham operation), the heart and liver were excised to perform histological and molecular biological analyses under general anesthesia. The hearts were removed, and the ventricles were dissected free of atrial tissue and large blood vessels. The RV was carefully separated from the left ventricle (LV) and the intraventricular septum. Fresh ventricular tissue was immediately blotted dry and weighed separately for determining the degree of RV hypertrophy based on two parameters: RV wall weight/biventricular weight (RV/BV), and RV wall weight/body weight (RV/BW). Fresh liver tissues were also immediately blotted dry and weighed for determining the degree of abdominal organ congestion owing to right heart failure: Liver weight/BW (Liver/BW).

The excised RV wall was fixed with either 10% buffered formalin for paraffin sections, or 4% paraformaldehyde for frozen sections. The paraffin sections of the RV were stained with picrosirius red for assessing the degree of myocardial fibrosis and were stained with Periodic Acid-Schiff for assessing myocardial cell size. The paraffin sections of the RV were used for immunohistochemistry, and were labeled using polyclonal CD31 antibody (1:50, Abcam, Cambridge, UK) and anti-α-smooth muscle actin antibody (1:50, DAKO, Hovedstaden, Denmark) for assessing capillary vascular density and vascular maturity. Frozen sections of the RV were also used for immunohistochemistry and were labeled with dihydroethidium for estimating superoxide production. The paraffin sections of the liver were stained with hematoxylin-eosin for assessing the degree of congestion around the central vein in the hepatic lobule.

Data were analyzed and averaged from three randomly selected fields in the RV free wall and the RV anterior and posterior hinge points in each group. The fibrotic area was calculated as the percentage of the myocardial area using Metamorph image analysis software (Molecular Devices, Inc., Downingtown, PA, USA). BZ-analysis software (Keyence, Tokyo, Japan) was used for measuring the capillary density and dihydroethidium positive dots. Myocardial cell size was determined by drawing point-to-point perpendicular lines across the cross-sectional area of the cell at the level of the nucleus. The results were expressed as the average diameter of 10 myocytes randomly selected from each selected field by Periodic Acid-Schiff staining using a fluorescence microscope (BZ-9000, Keyence, USA).

## Real-time polymerase chain reaction

The RV free wall region of the excised heart samples was later immersed in an RNA stabilization solution (RNAlater, Invitrogen, Carlsbad, CA, USA). The total RNA was isolated from the infarct-border area using the RNeasy Kit (Qiagen, Hilden, Germany), and was reverse-transcribed using the Omniscript reverse transcriptase (Qiagen) enzyme. Real-time PCR was performed using TaqMan Gene Expression Assay Master Mix (Applied Biosystems, Foster City, CA, USA) on a 7500 Fast Real-Time PCR System (Applied Biosystems). The following genes were analyzed using the TaqMan Gene Expression Assay (Applied Biosystems): *vascular*

*endothelial growth factor A*, *hepatocyte growth factor*, *C-X-C motif chemokine ligand 12*, *NADPH oxidase heavy chain subunit*, *NADPH oxidase 4*, *natriuretic peptide A*, *natriuretic peptide B*, *transforming growth factor beta 1*, and *cellular communication network factor 2*. *Glyceraldehyde-3-phosphate dehydrogenase* was co-amplified as an internal control for RNA integrity.

### Statistical analyses

All data were analyzed using the JMP Pro15 software (SAS Institute, Cary, NC, USA). All values were expressed as means ± standard deviation (SD). Comparisons between the two groups were performed using the Mann-Whitney U test. The differences were considered statistically significant at a P-value of <0.05.

## Results

### Amelioration of pressure-overloaded right ventricular dysfunction after patch implantation

The functional effect of autologous skeletal myoblast patches on the pressure-overloaded right ventricles was assessed using MDCT (Fig 2A) and TTE. The deterioration in RV volume, RVEF, and RV systolic and diastolic functions, at two months after PAB (before patch implantation) was not significantly different between the PATCH and the control groups (Fig 2B and 2C).

Two months after the patch implantation or sham operation, RV volume dilatation and reduction in RVEF were significantly alleviated in the PATCH group than in the control group (RVEDV: 47.5 ± 6.9 mL vs. 65.7 ± 7.4 mL, p = 0.01; RVESV: 27.8 ± 3.8 mL vs. 46.1 ± 5.2 mL, p <0.01; RVEF: 41.3 ± 1.1% vs. 29.7 ± 4.0%, p <0.01, PATCH vs. control, respectively) (Fig 2A).

In the TTE at two months after patch implantation or sham operation, the RV free wall thickness was not different in the groups (S1 Fig), but the RV systolic and diastolic functions were more significantly ameliorated in the PATCH group than in the control group (Tei index: 0.42 ± 0.08 vs. 0.68 ± 0.07, p <0.01; RVFAC: 47.8 ± 2.2% vs. 19.2 ± 5.2%, p <0.01; ICT: 46.0 ± 6.7 ms vs. 90.8 ± 9.8 ms, p <0.01; TAPSE: 9.0 ± 1.2 mm vs. 5.1 ± 0.9 mm, p<0.01; E/e': 6.57 ± 1.24 vs. 10.11 ± 1.38, p <0.01; IRT: 53.3 ± 7.8 ms vs. 99.7 ± 9.0 ms, p <0.01, PATCH vs. control, respectively) (Figs 2B and S1).

Pressure-volume loop analysis at two months after patch implantation and sham operation also revealed a more significant alleviation of the RV systolic and diastolic functions in the PATCH group than in the control group (Table 1 and S2 Fig). However, there was no difference in the systemic RV pressure, and the ratio between the systemic RV pressure and systemic LV pressure (Table 1), which indicates that the level of pressure load on the RV was equal between the groups.

### Significant improvement in myocardial oxidative metabolism in the patch implantation group

Based on the $^{11}$C-acetate PET conducted at 2 months after PAB (that is, before patch implantation or sham operation), and at 4 months after PAB (2 months after patch implantation or sham operation); the $MVO_2$ and MBF were measured at each point, and the preoperative values were subtracted from the postoperative values to evaluate the changes in the value between treatments (Fig 3).

The postoperative value of the $^{11}$C-acetate clearance rate constant k ($k_{mono}$), which is an index of tricarboxylic acid (TCA) cycle activity, was lower than the preoperative value at rest conditions, and showed no difference between the groups (Table 2). However, under stress conditions with dobutamine administration, the change in $k_{mono}$ was positive in the PATCH group but was negative in the control group (Table 2). The ratio between the $k_{mono}$ at stress and rest conditions,

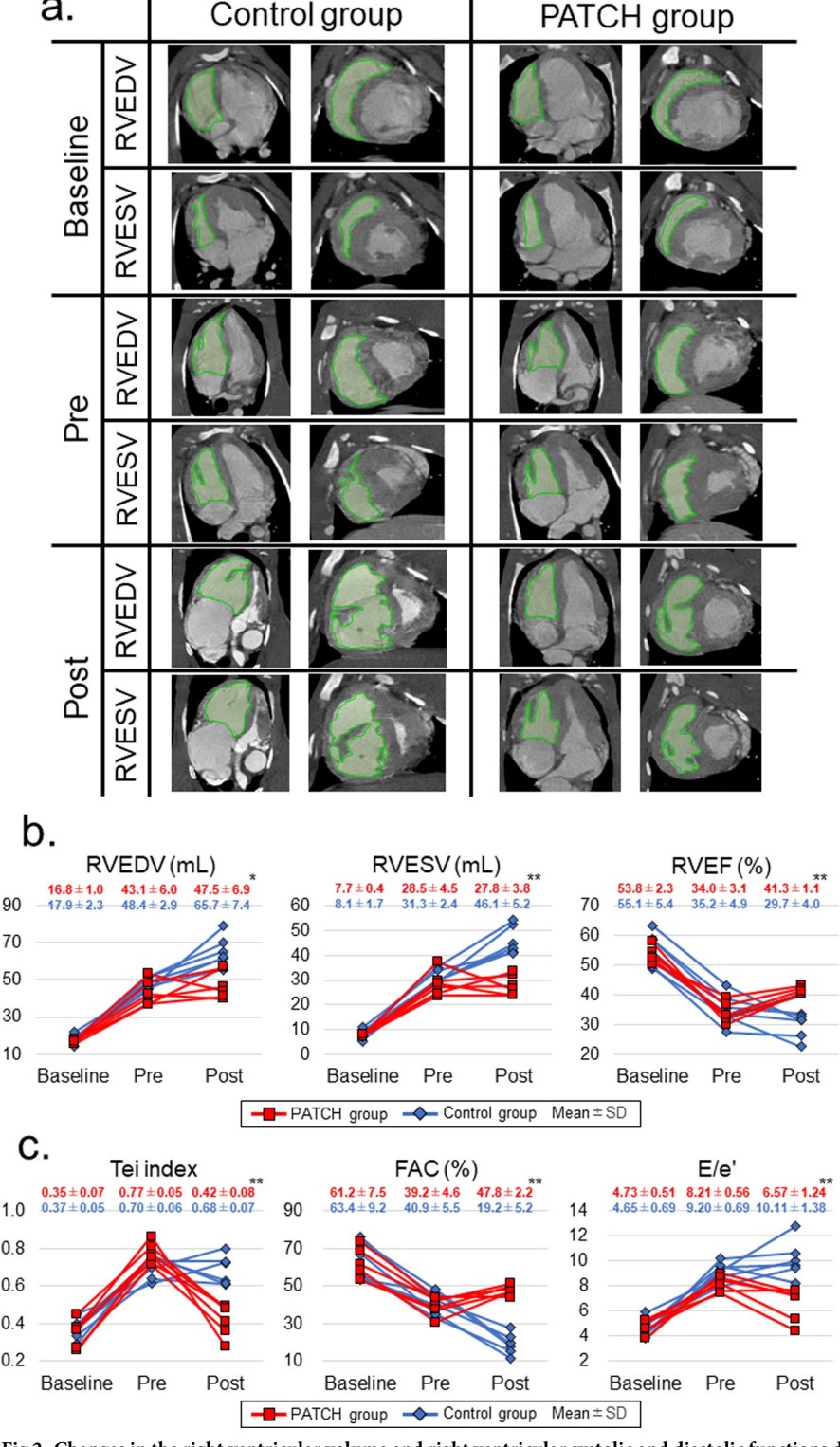

**Fig 2. Changes in the right ventricular volume and right ventricular systolic and diastolic functions during the study.** (a) Representative cardiac images of the four-chamber view and short axial view at the end-diastolic phase and end-systolic phase for each point. The RV volume was enlarged at the pre-treatment point compared to the baseline in either group, but at the post-treatment point, the enlargement of the RV volume was alleviated more in the PATCH group than in the control group. (b) Changes in the RV volume and RVEF at pre-treatment and post-treatment from

baseline. The RV volume dilatation and reduction of RVEF were more significantly alleviated in the PATCH group than in the control group at the post-treatment point. (c) Changes in the systolic and diastolic functions of the RV at pre-treatment and post-treatment, from the baseline. The RV systolic and diastolic functions significantly ameliorated in the PATCH group than in the control group at the post-treatment point. The red and blue numbers at each point represent the mean ± standard deviation of the PATCH group and the control group, respectively. P-values were calculated using the Mann-Whitney U test. P<0.05*, P<0.01** versus control group. RVEDV, right ventricular end-diastolic volume; RVESV, right ventricular end-systolic volume; RVEF, right ventricular ejection fraction; FAC, fraction area change; E/e', the ratio of early trans-tricuspid flow velocity to early diastolic velocity of the tricuspid annulus.

which is an index of the $MVO_2$ reserve, was positive in the PATCH group but was negative in the control group with a significant difference, which indicates a high activity reserve of the TCA cycle in the PATCH group (Table 2). RV efficiency calculated from RV work and RV oxygen consumption was significantly higher in the PATCH group than in the control group (Table 2).

Like the $MVO_2$, the postoperative value of the MBF was lower than the preoperative value at rest conditions and showed no difference between the groups (Table 2). However, under stress conditions with dobutamine administration, the change in MBF was positive in the PATCH group but was negative in the control group, with a significant difference (Table 2). The ratio between the MBFs at stress and rest conditions, which is an index of myocardial flow reserve, was positive in the PATCH group, but was negative in the control group with a significant difference, which indicates an increment of myocardial flow reserve after treatment in the PATCH group, but a decrease in the control group (Table 2).

## Myoblast cell patch attenuates pathological remodeling and myocardial ischemia in the pressure-overloaded right ventricular myocardium

The macropathological findings of the resected heart revealed a hypertrophied RV wall in both groups, however, the enlargement of the RV cavity was lesser in the PATCH group than in the

**Table 1. Analysis of hemodynamic indices at two months after patch implantation.**

|  |  | Control (n = 6) | PATCH (n = 6) | P-value |
|---|---|---|---|---|
| **Basic hemodynamic indices** |  |  |  |  |
|  | HR (bpm) | 75 ± 14 | 124 ± 26 | 0.013 |
|  | RVEF (%) | 23.7 ± 11.1 | 44.6 ± 9.6 | 0.008 |
|  | RVESP (mmHg) | 28.6 ± 10.2 | 35.9 ± 7.8 | 0.128 |
|  | RVESP/LVESP (mmHg) | 0.57 ± 0.17 | 0.75 ± 0.16 | 0.230 |
|  | RVEDP (mmHg) | 7.5 ± 1.3 | 2.9 ± 2.2 | 0.006 |
|  | dP/dt max (mmHg/sec) | 179 ± 55 | 425 ± 135 | 0.008 |
|  | dP/dt min (mmHg/sec) | −205 ± 76 | −391 ± 112 | 0.013 |
|  | Tau (ms) | 51.8 ± 11.4 | 32.2 ± 6.7 | 0.005 |
| **Load-independent parameters analyzed using PV loop** |  |  |  |  |
|  | ESPVR (mmHg/mL) | 1.15 ± 0.58 | 3.91 ± 1.54 | 0.005 |
|  | EDPVR (mmHg/mL) | 0.95 ± 0.39 | 0.14 ± 0.04 | 0.005 |

HR, heart rate; RVEF, right ventricular ejection fraction; RVESP, right ventricular end-systolic pressure; LVESP, left ventricular end-systolic pressure; RVEDP, right ventricular end-diastolic pressure; dP/dt max, the maximum rate of the right ventricular pressure rise; dP/dt min, the maximal rate of fall of the right ventricular pressure; PV, pressure-volume; Tau, the time constant of isovolumic relaxation; ESPVR, end-systolic pressure-volume relationship; EDPVR, end-diastolic pressure-volume relationship. P-values were calculated using the Mann-Whitney U test.

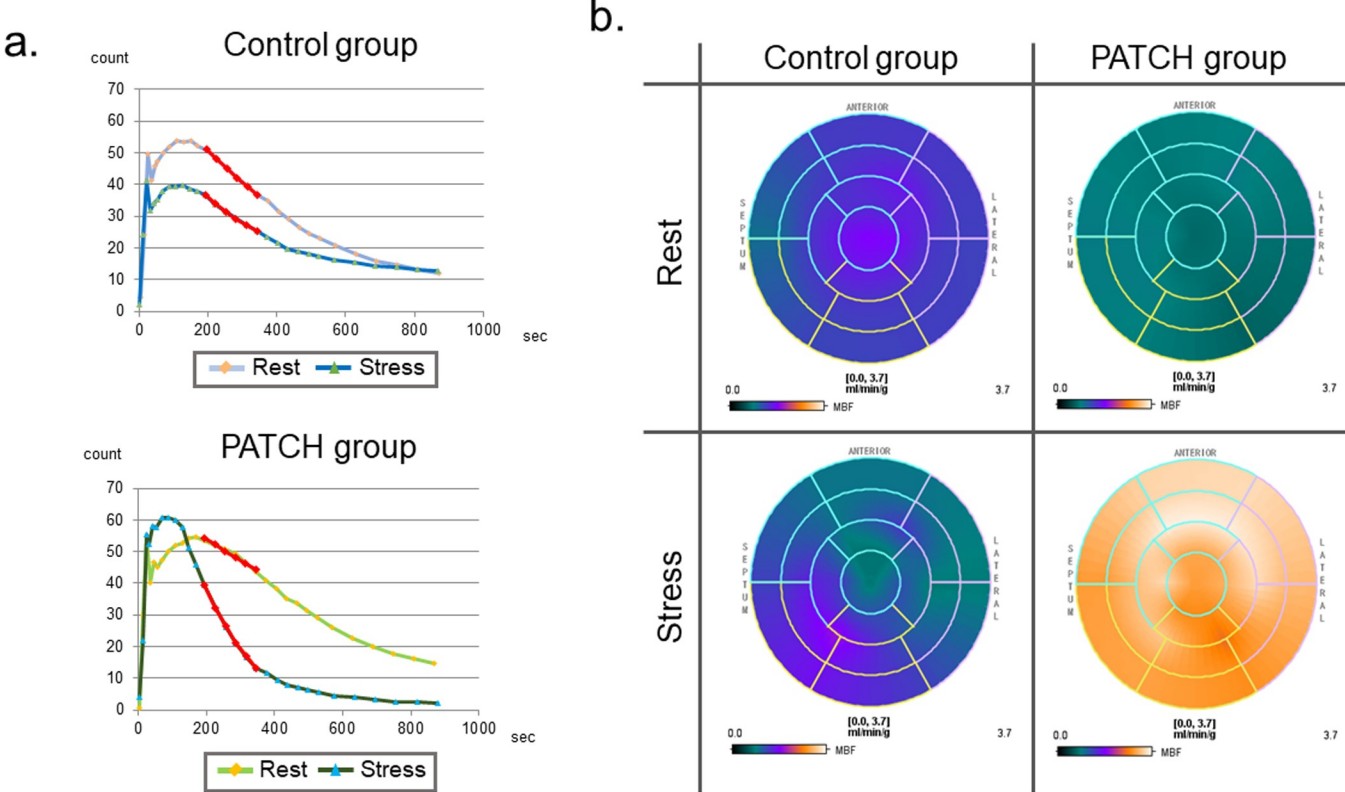

**Fig 3. Representative images of $^{11}$C-acetate clearance curve and a bullseye map of myocardial blood flow at two months after patch implantation or sham operation.** (a) The $^{11}$C-acetate clearance curve for stress conditions was steeper than the curve for rest conditions in the PATCH group, but the $^{11}$C-acetate clearance curve for stress conditions was flatter than the curve for rest conditions in the control group. (b) The bullseye map for stress conditions was globally upregulated from the bullseye map for rest conditions in the PATCH group, but the bullseye map for stress conditions showed no further remarkable change compared to the bullseye map for rest conditions in the control group. Rest, rest conditions; Stress, stress conditions with administration of dobutamine.

control group (Fig 4A). Additionally, the increase in RV mass was significantly lower in the PATCH group than in the control group for both the ratio of RV mass to biventricular mass, and the ratio of RV mass to the BW of the minipigs (Fig 4A).

**Table 2. Analysis of the change in myocardial oxidative metabolism and myocardial blood flow between treatments.**

|  |  | Control (n = 4) | PATCH (n = 4) | P-value |
|---|---|---|---|---|
| $k_{mono}$ |  |  |  |  |
|  | Rest conditions (min$^{-1}$, mean ± SD) | −0.003 ± 0.030 | −0.008 ± 0.015 | 0.659 |
|  | Stress conditions (min$^{-1}$, mean ± SD) | −0.078 ± 0.067 | 0.005 ± 0.050 | 0.194 |
|  | Stress/Rest (min$^{-1}$, mean ± SD) | −1.023 ± 0.990 | 0.185 ± 0.211 | 0.029 |
|  | RV efficiency (J/s·min$^{-1}$·g, mean ± SD) | 0.037 ± 0.022 | 0.311 ± 0.213 | 0.030 |
| Myocardial blood flow |  |  |  |  |
|  | Rest conditions (min$^{-1}$, mean ± SD) | −0.235 ± 0.331 | −0.128 ± 0.221 | 0.665 |
|  | Stress conditions (min$^{-1}$, mean ± SD) | −1.565 ± 1.569 | 0.625 ± 0.510 | 0.030 |
|  | Stress/Rest (min$^{-1}$, mean ± SD) | −1.323 ± 1.979 | 0.785 ± 0.463 | 0.005 |

$k_{mono}$, $^{11}$C-acetate clearance rate constant k; Stress/Rest, the ratio between the value of stress conditions and rest conditions; RV, right ventricular. The P-values were calculated using the Mann-Whitney U test.

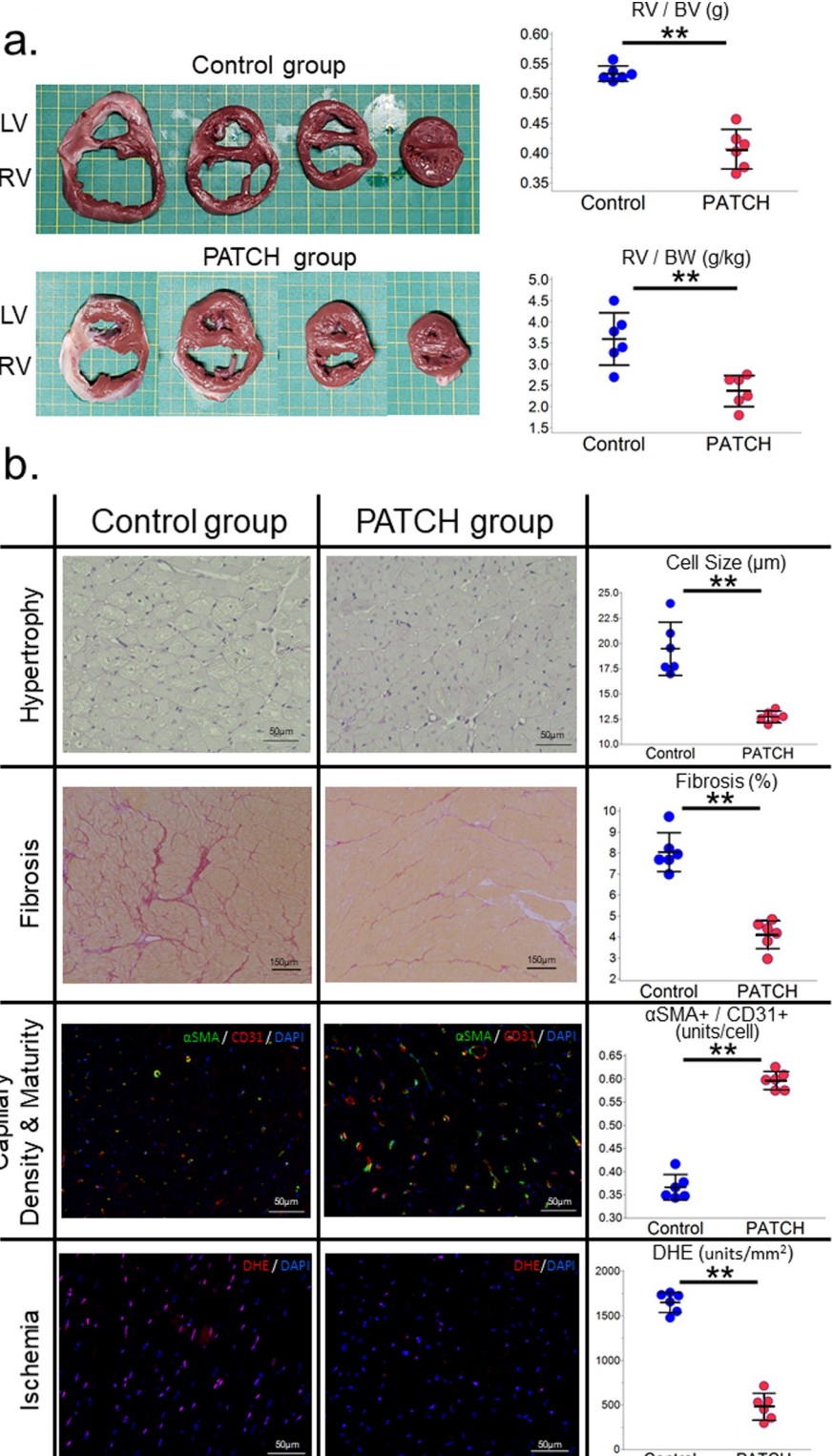

**Fig 4. Histopathological analysis of pressure-overloaded right ventricular myocardium two months after patch implantation or sham operation.** (a) Representative figures of the macropathological findings. Despite the hypertrophied RV wall, the enlargement of the RV cavity was well-controlled in the PATCH group when compared to the control group. The increase in RV mass was significantly lower in the PATCH group than in the control group for both the ratio between RV mass and biventricular mass, and the ratio between the RV mass and the BW of the

minipigs. (b) Representative photomicrographs of Periodic Acid-Schiff staining (×400, scale bar = 50 mm), picrosirius red staining (×100, scale bar = 150 mm), double immunohistochemical staining of anti-CD31 and anti-aSMA (×400, scale bar = 50 mm), and dihydroethidium staining (×400, scale bar = 50 mm). Cardiomyocyte size and fibrotic area were significantly smaller in the PATCH group than in the control group. The number of CD31 and aSMA double positive arterioles and capillaries per cardiomyocyte significantly increased in the PATCH group than in the control group. Superoxide production was significantly reduced in the PATCH group than in the control group. The horizontal line in the middle indicates the mean, and the whiskers mark indicates the standard deviation. P-values were calculated using the Mann-Whitney U test. P<0.01**. LV, left ventricle; RV, right ventricle; BV, bi ventricle; BW, bodyweight of mini-pigs; DHE, dihydroethidium staining. Histological analysis revealed that the cardiomyocyte size and fibrotic area were significantly smaller in the PATCH group than in the control group (cell size: 12.7 ± 0.5 μm vs. 19.5 ± 2.6 μm, p <0.01; fibrosis: 4.1 ± 0.7% vs. 8.0 ± 0.9%, p <0.01, PATCH vs. control, respectively) (Fig 4B). The capillary vascular density and vascular maturity were assessed by CD31 and aSMA double immunostaining. The number of CD31 and aSMA double positive arterioles and capillaries per cardiomyocyte significantly increased in the PATCH group when compared to the control group (0.60 ± 0.02 units/cell vs. 0.37 ± 0.03 units/cell, respectively, p <0.01) (Fig 4B). Superoxide production was assessed using dihydroethidium staining, and superoxide production was significantly lower in the PATCH group than in the control group (481 ± 149 units/mm$^2$ vs. 1,649 ± 114 units/mm$^2$, respectively, p <0.01) (Fig 4B).

For the pathological evaluation of the liver, to identify the adverse effects of RV failure on abdominal organs, the ratio between the liver mass and BW of minipigs was significantly smaller in the PATCH group than in the control group, and the congestion around the central vein in the hepatic lobule suggested that the degree of damage by RV failure was more remarkable in the control group when compared to the PATCH group (S3 Fig).

## Upregulated VEGF, HGF, and SDF-1, and downregulated NOX-2, NOX-4, ANP, BNP, TGFβ1, and CTGF in the heart after myoblast cell patch implantation

Real-time PCR was used to quantitatively assess the expression levels of myoblast cell patch-derived factors, such as vascular endothelial growth factor (VEGF), hepatocyte growth factor (HGF), and stromal cell-derived factor-1 (SDF-1), as well as to assess the expression levels of myocardial hypoxia, hypertrophy, and fibrosis (Fig 5). Intramyocardial mRNA levels of VEGF, HGF, and SDF-1 were more significantly upregulated in the PATCH group than in the control group (VEGF: 1.08 ± 0.36 vs. 0.78 ± 0.06, p = 0.02; HGF: 1.34 ± 0.75 vs. 0.78 ± 0.11, p = 0.02; SDF-1: 1.26 ± 0.83 vs. 0.49 ± 0.23, p<0.01, PATCH vs. control, respectively) (Fig 5A). Intramyocardial mRNA levels of NADPH Oxidase (NOX)-2, NOX-4, atrial natriuretic peptide (ANP), brain natriuretic peptide (BNP), transforming growth factor-β1 (TGF-β1), and cellular communication network factor 2 (CCN2) were significantly downregulated in the PATCH group than in the control group (NOX-2: 0.95 ± 0.30 vs. 2.21 ± 0.84, p<0.01; NOX-4: 0.75 ±0.15 vs. 2.01 ± 0.54, p<0.01; ANP: 0.47 ± 0.26 vs. 2.40 ± 1.82, p<0.01; BNP: 0.15 ± 0.11 vs. 5.26 ± 4.61, p<0.01; TGF-β1: 0.84 ± 0.13 vs. 1.34 ± 0.21, p<0.01; CCN2: 0.39 ± 0.25 vs. 1.36 ± 1.04, p = 0.02, PATCH vs. control, respectively) (Fig 5B).

## Discussion

In this study, we performed autologous skeletal myoblast patch implantation in pressure-overloaded right heart failure in a porcine model. Serial measurements of RV volume with MDCT revealed that the dilatation of the RV due to pressure overload could be attenuated by myoblast patch transplantation (Fig 2A and 2B). This was accompanied by the improvement in the systolic and diastolic functions of the RV, assessed on TTE (Fig 2C) or pressure-volume loop analysis by cardiac catheterization (Table 1), 2 months after the patch transplantation. In addition, the right ventricular myocardial oxidative metabolism and myocardial flow reserve, assessed with $^{11}$C-acetate PET, were significantly increased by myoblast patch transplantation

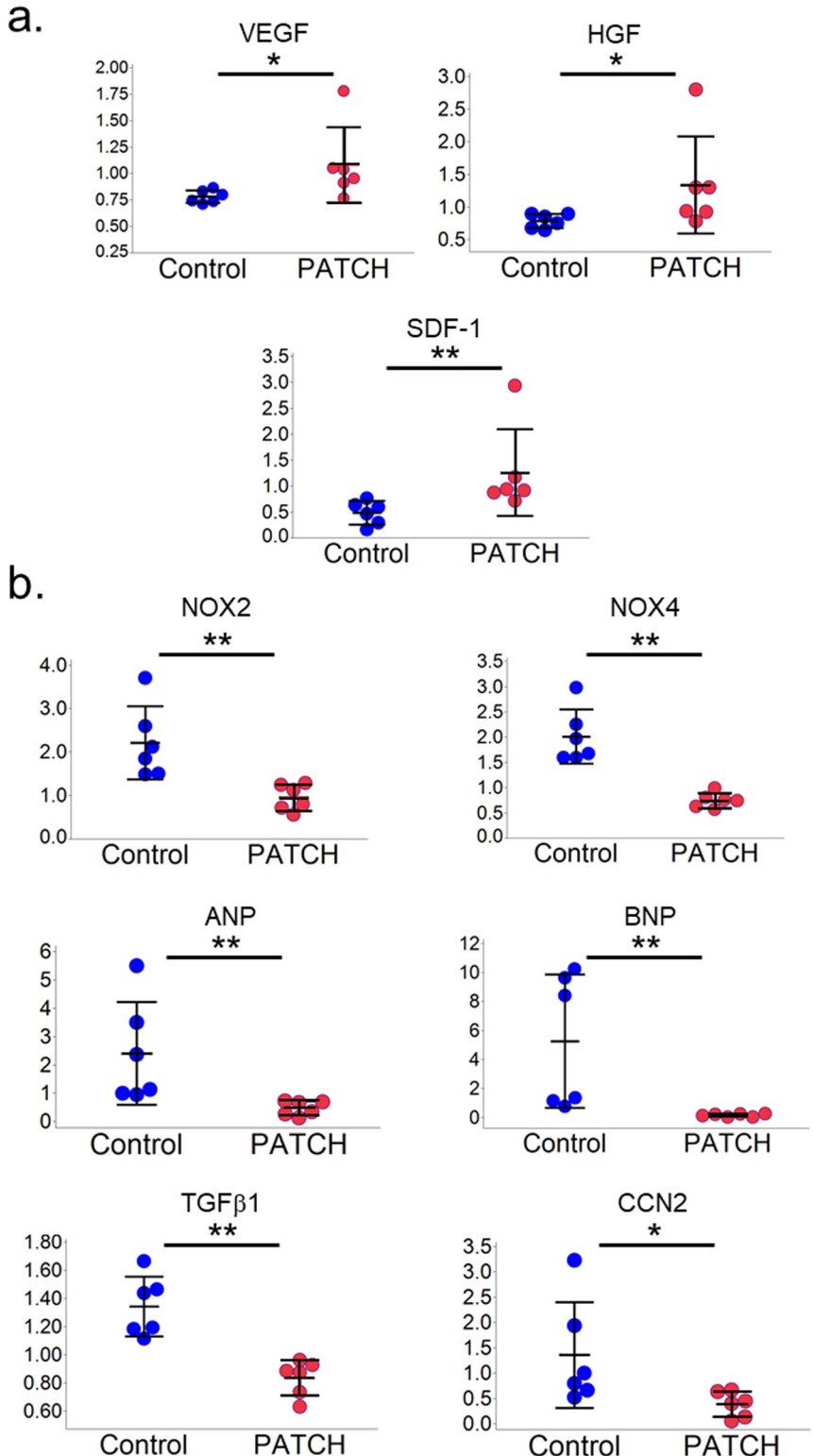

**Fig 5. Gene expression of neurohormonal factors in the heart tissue assessed using real-time PCR.** (a) Intramyocardial mRNA levels of VEGF, HGF, and SDF-1 were more significantly upregulated in the PATCH group than in the control group. (b) Intramyocardial mRNA levels of NOX-2, NOX-4, ANP, BNP, TGF-β1, and CCN2 were significantly downregulated in the PATCH group as compared to the control group. The horizontal line in the middle indicates the mean, and the whiskers mark indicates the standard deviation. The P-values were calculated using the

Mann-Whitney U test. $P<0.05^*$, $P<0.01^{**}$. VEGF, vascular endothelial growth factor; HGF, hepatocyte growth factor; SDF-1, stromal cell-derived factor-1; ANP, atrial natriuretic peptide; BNP, brain natriuretic peptide; TGF-β1, transforming growth factor-β1; CCN2, cellular communication network factor 2.

(Table 2). Histological analysis showed reduced myocardial hypertrophy, reduced fibrosis, and increased aSMA and CD31 double-positive mature vessels in the myocardium, by myoblast patch transplantation (Fig 4). Gene expression profiling showed a significant upregulation of angiogenic cytokines and reduced oxidative stress markers in the myocardium by myoblast patch transplantation (Fig 5). These results suggest that, the transplantation of a myoblast patch induces angiogenesis and anti-fibrosis by means of a cytokine paracrine effect, and results in therapeutic effects in pressure-overloaded RV failure.

There have been multiple cell transplantation preclinical studies for RV dysfunction with various cell types, animal models, and administration methods [44,45]. However, so far, none of them have achieved clinical application. In the present study, we used a pressure-overloaded right heart model to target CHD patients with a systemic right ventricle, which of right ventricle is exposed to a high systemic vascular resistance for a long period. We directly attached the myoblast cells to the epicardium of the RV myocardium with heart failure, as an administration method. Our cell source has already been applied clinically for left heart failure, where its safety has been assured and the detailed mechanism has been well studied. Furthermore, our cell patch transplantation can more reliably transfer therapeutic cells to the myocardium in heart failure, compared to previously reported administration methods, such as transcoronary or intramyocardial injections. Our cell patch technology also has the advantage of not only maintaining the cell-cell connection, but also noninvasively harvesting its own extracellular matrix, deposited during culture beneath the cell patch, which allows a greater anti-heart failure effect.

Histopathologically, the loss of capillaries in the RV myocardium is a well-known feature in cases of pressure-overloaded RV dysfunction in CHD patients with systemic RV, such as hypoplastic left heart syndrome with previous Norwood/Fontan palliation [46–48]. However, the $MVO_2$ was increased by hypertrophic changes in the ventricular cardiomyocytes in a pressure-overloaded right heart [43,49–51]. Under these conditions, the RV cardiomyocytes, which were originally poorly resistant to ischemia [8,12,13], were in seriously ischemic conditions. The metabolic remodeling in cardiomyocytes owing to increased oxidative stress, due to the ischemic conditions of a pressure-overloaded RV, can cause various reactions, such as inflammation, apoptosis, and fibrosis, leading to RV dysfunction [8,12,13]. Based on these conditions of the pressure-overloaded RV myocardium, the main therapeutic mechanism in autologous skeletal myoblast patch implantation therapy—angiogenesis—may be effective for pressure-overloaded right heart failure.

The angiogenic effect of autologous skeletal myoblast patch implantation therapy has been previously proven in many of our studies on left heart failure [33,40]. The major underlying mechanism, angiogenesis, is expected to be upregulated by pro-angiogenetic cytokines from the myoblast patch, such as VEGF and HGF [34,52,53]. These cytokines act on vascular endothelial cells and vascular smooth muscle cells to promote angiogenesis along with vascular maturity of the myocardium in a pressure-overloaded right heart. Additionally, the upregulated SDF-1 expression from the patch may promote further angiogenesis, secondary to the accumulation of mesenchymal stem cells from the bone marrow on the impaired myocardium of the right ventricle [34,52–55]. In our present study, the upregulation of VEGF, HGF, and SDF-1 was observed, and the same mechanism of action is expected to work on the pressure-overloaded right ventricular myocardium. In the present study, we did not investigate the survival of the implanted myoblast cells. However, we confirmed the survival of the implanted myoblast cells for a few months

after transplantation in our previous study on left heart failure [35,56,57], and the same survival can be expected in the right ventricle. Even after the loss of implanted myoblast cells a few months after implantation, the recruited bone marrow mesenchymal stem cell-derived vascular endothelial cells expressed angiogenic cytokines, along with the maintenance or maturation of newly developed vascular networks. Therefore, the functional recovery of heart failure is preserved even after cell loss. Moreover, as in myoblast patch treatment for left heart failure, the anti-hypertrophic effect on cardiomyocytes in the pressure-overloaded RV decreased $MVO_2$ and ameliorated myocardial dysfunction, by improving the myocardial ischemia.

In the initial stage of pressure-overloaded right heart failure, compensatory mechanisms such as volume dilatation and hypertrophic change work to maintain the RV function in a hyperdynamic state [8,12,13]. In this stage, the myocardial oxidative metabolism improves, and coronary flow reserve can be secured [43,49–51,58,59]. However, the prolonged pressure overload advances the metabolic remodeling of cardiomyocytes, owing to persistent mechanical stress on the myocardium and severe ischemic conditions [8,12,13,19,20]. Eventually, the right heart failure progresses to a decompensated stage with impaired RV function due to the reduction in cardiac reserve [13,60–62]. Considering the results of the [11]C-acetate PET, the angiogenic effect of patch implantation suppresses the metabolic remodeling of cardiomyocytes and maintains the cardiac reserve of the RV, by avoiding chronic ischemic conditions. Thus, patch implantation prevents progression to a decompensated stage and maintains a hyperdynamic state of compensatory right heart failure. Additionally, from the results of cardiac function analysis, the suppressed ventricular volume dilatation and decreased end-diastolic pressure with patch implantation suggest a reduction in the RV wall stress, which can also work positively for the amelioration of right heart failure.

## Conclusions

In summary, in pressure-overloaded right heart porcine models, autologous skeletal myoblast patch implantation therapy alleviated myocardial ischemia by angiogenesis and improved the myocardial oxidative metabolism as well as the myocardial blood flow. This led to the amelioration of RV diastolic and systolic functions, thereby alleviating right heart failure. This study demonstrated the potential of autologous skeletal myoblast patch implantation therapy as a novel cardiac regenerative treatment for patients with right heart failure associated with CHD.

## Supporting information

**S1 Fig. Changes in the RV systolic and diastolic functions during the study.** RV systolic and diastolic functions were significantly ameliorated in the PATCH group than in the control group, at two months after patch implantation or sham operation. The red and blue numbers at each point represent the means ± standard deviation of the PATCH group and control group, respectively. P-values were calculated using the Mann-Whitney U test. $P<0.01^{**}$ versus control group. ICT, isovolumic contraction time; TAPSE, tricuspid annular plane systolic excursion; IRT, isovolumic relaxation time; RVFWT, right ventricular free wall thickness.
(TIF)

**S2 Fig. Representative pressure–volume loops of the control and PATCH groups under different loading conditions.** The slope of the end-systolic pressure–volume relationship is displayed as a black straight line on top of the pressure-volume loops. The correlation of the end-diastolic pressure–volume relationship is displayed as a back straight line below the pressure-volume loops. RVP, right ventricular pressure; RV, right ventricle.
(TIF)

**S3 Fig. Adverse effects of RV failure on the abdominal organs.** (a) Representative photomicrographs of hematoxylin-eosin staining (×40, scale bar = 300 mm; ×100, scale bar = 100 mm). Congestion around the central vein in the hepatic lobule was more marked in the control group than in the PATCH group. (b) The ratio between the liver mass and the bodyweight of minipigs was significantly lower in the PATCH group than in the control group. The horizontal line in the middle indicates the mean, and the whiskers mark indicates the standard deviation. P-values were calculated using the Mann-Whitney U test. P<0.05*.
(TIF)

**S1 File.**
(XLSX)

## Acknowledgments

We thank Kaori Ikuma for her expert assistance with real-time PCR and Isamu Matsuda, Ooyama Kenji, and Fumiya Oohasi for their technical support in creating the skeletal myoblast cell patches.

## Author Contributions

**Conceptualization:** Kanta Araki, Shigeru Miyagawa, Takuji Kawamura, Yoshiki Sawa.

**Data curation:** Kanta Araki, Ryo Ishii, Tadashi Watabe, Akima Harada.

**Formal analysis:** Kanta Araki, Takuji Kawamura, Tadashi Watabe.

**Funding acquisition:** Kanta Araki, Shigeru Miyagawa, Yoshiki Sawa.

**Investigation:** Kanta Araki.

**Methodology:** Kanta Araki, Shigeru Miyagawa, Takuji Kawamura, Takayoshi Ueno, Yoshiki Sawa.

**Project administration:** Kanta Araki.

**Resources:** Kanta Araki.

**Supervision:** Shigeru Miyagawa, Takuji Kawamura, Masaki Taira, Koichi Toda, Toru Kuratani, Takayoshi Ueno, Yoshiki Sawa.

**Validation:** Kanta Araki, Takuji Kawamura, Ryo Ishii, Tadashi Watabe.

**Visualization:** Kanta Araki, Akima Harada.

**Writing – original draft:** Kanta Araki.

**Writing – review & editing:** Shigeru Miyagawa, Takuji Kawamura, Yoshiki Sawa.

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
