## [Decision Letter · Decision Letter 0]

11 Jan 2021

PONE-D-20-39094

Autologous skeletal myoblast patch implantation prevents the deterioration of myocardial ischemia and right heart dysfunction in a pressure-overloaded right heart porcine model

PLOS ONE

Dear Dr. Araki,

Thank you for submitting your manuscript to PLOS ONE. After careful consideration, we feel that it has merit but does not fully meet PLOS ONE’s publication criteria as it currently stands. Therefore, we invite you to submit a revised version of the manuscript that addresses the points raised during the review process.

We look forward to receiving your revised manuscript.

Kind regards,

Michiya Matsusaki

Academic Editor

PLOS ONE

Journal Requirements:

Reviewers' comments:

Reviewer's Responses to Questions

**Comments to the Author**

1. Is the manuscript technically sound, and do the data support the conclusions?

Reviewer #1: Yes

Reviewer #2: Yes

2. Has the statistical analysis been performed appropriately and rigorously? 

Reviewer #1: Yes

Reviewer #2: Yes

3. Have the authors made all data underlying the findings in their manuscript fully available?

Reviewer #1: Yes

Reviewer #2: Yes

4. Is the manuscript presented in an intelligible fashion and written in standard English?

Reviewer #1: Yes

Reviewer #2: Yes

5. Review Comments to the Author

Reviewer #1: The authors demonstrated improvement of right ventricular dysfunction in pig model by transplanting autologous myoblast sheets. They made the pig model whose pulmonary artery was banded and fabricated the autologous graft by cell sheet-based tissue engineering. Right ventricular volume and ejection fraction were improved in the cell sheet transplantation group than in the control. Blood vessel formation, secretion of cytokines including VEGF, HGF and SDF1 were increased and metabolism in the heart was also improved in the cell sheet transplantation group. They concluded that autologous myoblast sheets should contribute to the future treatment of congenital heart diseases.

It is the advance that cell sheet transplantation is effective for right ventricular dysfunction in addition to left ventricular dysfunction. Some additional information will strengthen the work.

　How about the fate of transplanted myoblast sheets? Are there any data which indicate their survival? At least please discuss about the point in comparison with the previous papers.

What congenital diseases will be targets and how to transplant? It may be informative for readers.

Ln 92 “patchto” may be a typo.

Reviewer #2: The authors investigated the treatment efficacy of autologous skeletal myoblast patch for the right ventricular dysfunction using a pressure-overload porcine model. They observed significant amelioration of right ventricular dysfunction with improved oxidative metabolism and myocardial flow reserve. Furthermore, histopathological and gene expression analysis confirmed improved myocardial ischemia by angiogenesis. From the standpoint of the intractability of right ventricular dysfunction, their findings will be of interest.

My comments are as below.

#1The data of each individual pig of Figures 2b, 2c, S1, and so on should be plotted or shown in the manuscript as well as the average data of each group.

#2 Cell transplantation studies for the treatment of right ventricular dysfunction previously reported should be referred and discussed in the paper. (e.g. Lambert et al. Journal of Thoracic and Cardiovascular Surgery, (2015), 708-715.e1, 149(3) )

#3 How long did transplanted myoblasts remain alive? The authors should investigate and mention whether or not any evidence of survival of the transplanted cells were observed in the imaging and histopathological analyses.

---

## [Author Response · Author response to Decision Letter 0]

19 Jan 2021

January 19, 2021

Michiya Matsusaki

Academic Editor

PLOS ONE

Dear Editor:

We wish to re-submit the manuscript titled “Autologous skeletal myoblast patch implantation prevents the deterioration of myocardial ischemia and right heart dysfunction in a pressure-overloaded right heart porcine model.” The manuscript ID is PONE-D-20-39094.

We thank you and the reviewers for your thoughtful suggestions and insights. The manuscript has benefited from these insightful suggestions. I look forward to working with you and the reviewers to move this manuscript closer to publication in the PLOS ONE.

The manuscript has been rechecked and the necessary changes, which are indicated with yellow highlights, have been made in accordance with the reviewers’ suggestions. The responses to all comments have been prepared and given below. 

Reviewer #1:

1) How about the fate of transplanted myoblast sheets? Are there any data which indicate their survival? At least please discuss about the point in comparison with the previous papers.

Response: Thank you for the insightful comment and suggestion. Based on preliminary data, we could confirm the engraftment of transplanted myoblast cells on the right ventricle one week after the transplantation [data not shown in the manuscript]. Thus, we estimated that the myoblast cells in this study could survive more than one week after transplantation. In addition, we observed the survival of transplanted myoblast cells on the left ventricle at one month after the transplantation, in our previous study on left heart failure [35,56]. Based on these studies, the transplanted myoblast cells can survive for few months after the transplantation while maintaining the therapeutic effects. Taken together, the myoblast cells transplanted onto the right ventricle in this study could also survive for a few months after the transplantation to maintain the therapeutic effects.

We revised the manuscript as follows, adding our previous papers in the reference: “In the present study, we did not investigate the survival of the implanted myoblast cells. However, we confirmed the survival of implanted myoblast cells for a few months after transplantation in our previous study on left heart failure [35,56,57], and the same survival can be expected in the right ventricle.” (page 29-30, line 425-428).

2) What congenital diseases will be targets and how to transplant? It may be informative for readers.

Response: Thank you for the meaningful suggestion. In the present study, we used a pressure-overloaded right heart model to target congenital heart disease not only with pulmonary hypertension or right ventricular outflow tract obstruction, such as repaired tetralogy of Fallot, but also with a systemic right ventricle, such as hypoplastic left heart syndrome after Fontan procedure, complete transposition of the great arteries with previous atrial switch repair, and congenitally corrected transposition of the great arteries. In these congenital heart diseases with a systemic right ventricle, the right ventricle is exposed to a high systemic vascular resistance for a long period, resulting in the development of right ventricular dysfunction in the late phase, which is a predictor for poor prognosis. Myocardial regeneration therapy for right ventricular dysfunction will become more important due to an increasing number of congenital heart disease patients with right heart failure, as the outcomes of congenital heart disease improve. Although it is impossible to create an exact animal model of a systemic right ventricle, we believe our pressure-overloaded right heart porcine model is a reasonable surrogate for congenital heart disease with systemic right ventricle to determine the therapeutic effect. As for the timing of cell transplantation, we are considering patch transplantation as a concomitant procedure for surgical intervention at late phase, such as valve plasty or replacement.

We revised the manuscript as follows: “In the present study, we used a pressure-overloaded right heart model to target CHD patients with a systemic right ventricle, which of right ventricle is exposed to a high systemic vascular resistance for a long period. We directly attached the myoblast cells to the epicardium of the RV myocardium with heart failure, as an administration method. (page 27, line 392-394).

3) Ln 92 “patchto” may be a typo.

Response: Thank you for your suggestion. As you mentioned, “patchto” was a typo. We have corrected the manuscript as follows: “Additionally, some fibrin glue was applied over the cell patches after their placement to attach them to the epicardium of the heart.” (page 7, line 91-92).

Reviewer #2:

1) The data of each individual pig of Figures 2b, 2c, S1, and so on should be plotted or shown in the manuscript as well as the average data of each group.

Response: Thank you for the meaningful suggestion. As per your suggestion, we have revised the graphs in Figures 2b, 2c, 4, 5, S1, and S3b to visualize the data for each individual pig.

2) Cell transplantation studies for the treatment of right ventricular dysfunction previously reported should be referred and discussed in the paper.

Response: Thank you for the insightful suggestion. As you mentioned, there are many cell transplantation preclinical studies for right ventricular dysfunction with various cell types, animal models, and administration methods. However, none of them have achieved clinical application until now. The cell source used in this study has already been approved by Japanese Ministry of Health, Labour and Welfare, and applied clinically for left heart failure. Its safety has been assured, and the detailed mechanism has been well studied. Furthermore, our cell patch transplantation can more reliably transfer therapeutic cells to failure myocardium compared to previously reported administration methods, such as transcoronary or intramyocardial injection. Our cell patch technology also has the advantage of not only maintaining a cell-cell connection but also noninvasively harvesting its own extracellular matrix deposited during culture beneath the cell patch, which allows a greater anti-heart failure effect.

Therefore, we revised as follows: “There have been multiple cell transplantation preclinical studies for RV dysfunction with various cell types, animal models, and administration methods [44,45]. However, so far, none of them have achieved clinical application. In the present study, we used a pressure-overloaded right heart model to target CHD patients with a systemic right ventricle, which of right ventricle is exposed to a high systemic vascular resistance for a long period. We directly attached the myoblast cells to the epicardium of the RV myocardium with heart failure, as an administration method. Our cell source has already been applied clinically for left heart failure, where its safety has been assured and the detailed mechanism has been well studied. Furthermore, our cell patch transplantation can more reliably transfer therapeutic cells to the myocardium in heart failure, compared to previously reported administration methods, such as transcoronary or intramyocardial injections. Our cell patch technology also has the advantage of not only maintaining the cell-cell connection, but also noninvasively harvesting its own extracellular matrix, deposited during culture beneath the cell patch, which allows a greater anti-heart failure effect.” (pages 27–28, lines 390–402).

3) How long did transplanted myoblasts remain alive? The authors should investigate and mention whether or not any evidence of survival of the transplanted cells were observed in the imaging and histopathological analyses.

Response: Thank you for the insightful comment and suggestion. Based on preliminary data, we could confirm the engraftment of transplanted myoblast cells on the right ventricle one week after the transplantation [data not shown in the manuscript]. Thus, we estimated that the myoblast cells in this study could survive more than one week after transplantation. In addition, we observed the survival of transplanted myoblast cells on the left ventricle at one month after the transplantation, in our previous study on left heart failure [35,56]. Based on these studies, the transplanted myoblast cells can survive for few months after the transplantation while maintaining the therapeutic effects. Taken together, the myoblast cells transplanted onto the right ventricle in this study could also survive for a few months after the transplantation to maintain the therapeutic effects.

We revised the manuscript as follows, adding our previous papers in the reference: “In the present study, we did not investigate the survival of the implanted myoblast cells. However, we confirmed the survival of implanted myoblast cells for a few months after transplantation in our previous study on left heart failure [35,56,57], and the same survival can be expected in the right ventricle.” (page 29-30, line 425-428).

We thank you once again for the opportunity to enhance our manuscript with your valuable comments and queries. We have worked hard to incorporate your feedback, and we hope that these revisions persuade you to accept our submission. I look forward to hearing from you.

Sincerely,

Dr Yoshiki Sawa

Department of Cardiovascular Surgery, Osaka university graduate school of medicine 

2-2 Yamadaoka, Suita, 565-0871, Japan

Tel.: 81-6-6879-3154

FAX: 81-6-6879-3163

E-mail: sawa‑p@surg1.med.osaka‑u.ac.jp

---

## [Decision Letter · Decision Letter 1]

8 Feb 2021

Autologous skeletal myoblast patch implantation prevents the deterioration of myocardial ischemia and right heart dysfunction in a pressure-overloaded right heart porcine model

PONE-D-20-39094R1

Dear Dr. Araki,

We’re pleased to inform you that your manuscript has been judged scientifically suitable for publication and will be formally accepted for publication once it meets all outstanding technical requirements.

Kind regards,

Michiya Matsusaki

Academic Editor

PLOS ONE

Additional Editor Comments (optional):

Reviewers' comments:

Reviewer's Responses to Questions

**Comments to the Author**

1. If the authors have adequately addressed your comments raised in a previous round of review and you feel that this manuscript is now acceptable for publication, you may indicate that here to bypass the “Comments to the Author” section, enter your conflict of interest statement in the “Confidential to Editor” section, and submit your "Accept" recommendation.

Reviewer #1: All comments have been addressed

Reviewer #2: All comments have been addressed

---

## [Editor Report · Acceptance letter]

18 Feb 2021

PONE-D-20-39094R1 

Autologous skeletal myoblast patch implantation prevents the deterioration of myocardial ischemia and right heart dysfunction in a pressure-overloaded right heart porcine model 

Dear Dr. Araki:

I'm pleased to inform you that your manuscript has been deemed suitable for publication in PLOS ONE. Congratulations! Your manuscript is now with our production department. 

Kind regards, 

on behalf of

Dr. Michiya Matsusaki 

Academic Editor

PLOS ONE